# AICAR attenuates postoperative abdominal adhesion formation by inhibiting oxidative stress and promoting mesothelial cell repair

Yunhua Wu[1], Xianglong Duan[1,2], Zengzhan Gao[1], Ni Yang[1], Fei Xue[1,2]*

**1** The Second Department of General Surgery, Shaanxi Provincial People's Hospital, Xi'an, Shaanxi, China,
**2** Affiliated Hospital of Northwestern Polytechnical University, Xi'an, Shaanxi, China

* xfx1129@126.com

## Abstract

### Background

Postoperative abdominal adhesion is one of most common complications after abdominal operations. 5-aminoimidazole-4-carboxyamide ribonucleoside (AICAR) is an adenosine 5'-monophosphate activated protein kinase (AMPK) pathway agonist that inhibits inflammation, reduces cell fibrosis and cellular reactive oxygen species (ROS) injury, promotes autophagy and mitochondrial function. This study aimed to explore the mechanism of AICAR in inhibiting adhesion formation.

### Materials and methods

Forty rats were randomly divided into five groups. All of the rats except the sham group received cecal abrasion to establish an adhesion model. The rats in the sodium hyaluronate group were treated with 2 mL sodium hyaluronate before closing the peritoneal cavity. The AICAR 1 and 2 groups were treated with 100 mg/kg and 200 mg/kg AICAR, respectively. Seven days after the operation, all of the rats were euthanized, and the adhesion condition was evaluated by Nair's system. Inflammation was assessed by Eosin-hematoxylin (HE) staining and transforming growth factor-β (TGF-β1) detection. Oxidative stress effect was determined by ROS, nitric oxide (NO) level, superoxide dismutase (SOD), catalase, glutathione peroxidase (Gpx) and malondialdehyde (MDA) levels in adhesion tissue. Then, Sirius red picric acid staining was used to detect the fiber thickness. Immunohistochemical staining of cytokeratin-19 (CK-19), alpha-smooth muscle actin (α-SMA) and nuclear factor erythroid 2-related factor 2 (Nrf2) was also performed. Finally, HMrSV5 cells were treated with TGF-β1 and AICAR, the mRNA expression of E-cadherin, α-SMA and vimentin was assessed by q-PCR and cellular immunofluorescent staining.

### Results

The rats in the AICAR-treated group had fewer adhesion formation incidences and a reduced Nair's score. The inflammation was determined by HE staining and TGF-β1 concentration. The ROS, SOD, Catalase, Gpx, MDA levels and fiber thickness were decreased

**Data Availability Statement:** All relevant data are within the paper and its Supporting Information files.

**Funding:** Fei Xue received Postdoctoral Research Foundation of China (2022M712598); Fei Xue received 2021 Science and technology Talents Support project of Shaanxi Provincial People's Hospital (2021JY-08), Yunhua Wu received 2021 Science and technology Talents Support project of Shaanxi Provincial People's Hospital (2021JY-15). The funders had no role in study design, data collection and analysis, decision to publish, or preparation of the manuscript.

**Competing interests:** The authors have declared that no competing interests exist.

by AICAR treatments compared to the control. However, the NO production, Nrf2 levels and peritoneal mesothelial cell integrity were promoted after AICAR treatments. In vitro work, AICAR treatments reduced E-cadherin, α-SMA and vimentin mRNA level compared to that in the TGF-β1 group.

## Conclusion

AICAR can inhibit postoperative adhesion formation by reducing inflammation, decreasing oxidative stress response and promoting peritoneal mesothelial cell repair.

## 1 Introduction

As a most commonly complication after surgery, postoperative abdominal adhesion occured in 90%-95% of patients who underwent abdominal surgery will have abdominal adhesion formation, and 20% of these patients will suffer adhesion-related disease [1]. Despite surgical removal of the adhesion bands, there is currently no other effective treatment [2]. Therefore, preventing postoperative adhesion formation is very important.

The formation of postoperative adhesion is a complex process [3, 4]. After surgery, the injury or trauma of peritoneal tissue will lead to inflammation and collagen formation or deposition. Oxidative stress injury plays a vital role, and overaccumulation of superoxide not only harms mesothelial cells but also promotes cell transformation into mesenchymal cells [5]. Inhibiting reactive oxygen species (ROS) is a promising way to prevent adhesion formation [6]. Adenosine 5'-monophosphate activated protein kinase (AMPK) is an important pathway that maintains energy homeostasis and inhibits tissue injury [7]. A previous study illustrated that the AMPK pathway has a protective effect in chronic inflammatory diseases and can inhibit cell fibrosis, reduce ROS injury, promote autophagy, and improve mitochondrial function [8–10]. However, the function of the AMPK pathway in abdominal formation has not yet been determined.

The formation of abdominal adhesion is a process of inflammation that in turn leads to collagen deposition or fiber formation; thus, the AMPK pathway may have a protective effect by preventing adhesion in postoperative abdominal adhesion. Here, we demonstrated that the AMPK agonist 5-aminoimidazole-4-carboxyamide ribonucleoside (AICAR) [11] can reduce adhesion formation by inhibiting oxidative stress and promoting peritoneal repair in a rat model.

## 2 Materials and methods

The rats were purchased from the Experimental Animal Center of Xi'an Jiaotong University and kept in an approximately 22±2˚C environment. All rats had free access to food and water. The experiment was approved by the Ethics Committee of Xi'an Jiaotong University (No. XJTULAC2021-318).

### 2.1 Abdominal adhesion animal model

Forty Sprague–Dawley (SD) rats, aged＞3months and weighing 200g-250g, were used to establish the abdominal adhesion animal model as described in a our previous study. One day before surgery, the abdominal hair was removed, and fasting water was provided 6 hours before surgery. The rats were anesthetized under isoflurane anesthesia (2% isoflurane with 0.4

L/min $O_2$ flow). After skin sterilization, a 2- to 3-centimeter incision was created in the middle of the abdomen. Both the right side and cecum were rubbed with dry gauze 30 times until needle-like bleeding points appeared. Then, the injured cecum side was placed on the opposite side of the injured abdominal wall. Finally, the abdominal wall was closed by an intermittent suture in two layers.

## 2.2 Study design and adhesion assessment

Forty rats were randomly divided into five groups. The rats in the sham group received only open and close abdominal wall surgery, and the rats in the other groups underwent the abdominal adhesion model procedure. For the sodium hyaluronate group, 2 mL sodium hyaluronate was spread on the injured side of the abdominal wall and cecum before closing the abdominal wall. Within 7 days after surgery, the control group was treated with 1 mL normal saline through intraperitoneal injection on the left upper side of the abdomen, while AICAR Group 1 and Group 2 were treated with AICAR (#9944p, cell signaling technology, USA, purity>99%) 100 mg/kg and 200 mg/kg via subcutaneous injection, respectively, every day after surgery. Seven days after surgery, the rats were killed, and the adhesion condition was classified accoding to the Nair's score system, described as follows: Grade 0: complete absence of adhesions; Grade 1: single band of adhesion between viscera or from one viscera to the abdominal wall; Grade 2: two bands either between viscera or from viscera to the abdominal wall; Grade 3: more than two bands between viscera or from viscera to the abdominal wall, Grade 4: multiple dense adhesions or viscera directly adherent to the abdominal wall, irrespective of number or extent of adhesive bands [12]. Then, the adherent tissue was collected and divided into two pieces, one kept at -80˚C and the other fixed with 4% paraformaldehyde.

## 2.3 Hematoxylin-eosin (HE) staining and inflammation evaluation

For HE staining, the tissues were first fixed in 4% paraformaldehyde overnight and then made into 4 μm thick serial paraffin sections. HE staining was performed according to instructions of the HE staining kit (Beijing Soleibao Technology Co., Ltd., Beijing, China). The inflammation histopathological score was evaluated by the following system: 0 for the tissue without inflammatory cells; 1 for cases with observable macrophages, lymphocytes, and plasma cells; 2 for cases with macrophages, plasma cells, eosinophils, and neutrophils; and 3 for cases with inflammatory cell infiltration and microabscess formation. This process was performed by two independent researchers. At least five randomly selected high-power fields were reviewed for each section, at least four sections per rat were evaluated, and the score per section was the mean score of all sections per rat.

## 2.4 ELISA

An enzyme-linked immunosorbent assay (ELISA) was performed to detect the level of TGF-β1 production (CSB-E04727r, Shanghai Ximei Chemical Co. Ltd., Shanghai, China) and NO production (EMSNO, Invitrogen, Thermo Fisher Scientific, Inc.) according to the manufacturers' manual.

## 2.5 Chemiluminescence test for ROS

ROS were measured in the peritoneal lavage fluid 7 days after the operation. The lavage fluids were collected before opening the abdominal cavity. A commercial ROS detection kit (Sinovac Biochemical Reagents, Shanghai, China) was used according to the manufacturer's protocol. The concentrations of the samples were calculated using a standard curve.

## 2.6 Malondialdehyde (MDA) detection

The MDA concentration in the tissues was measured by an MDA detection kit (S0131S, beyotime biotechnology Co. Ltd., Shanghai, China) according to the manufacturer's instructions.

## 2.7 Superoxide dismutase (SOD), Catalase and Glutathione peroxidase (Gpx) activity detection

SOD activity was measured spectrophotometrically according to the methods of Flohe and Otting [13]. In this method, inhibition of the cytochrome *c* reduction rate is monitored at 550nm at 25˚C utilizing the xantine/xantine oxidase system as the source of $O_2$. SOD competes for superoxide and decreases the reduction rate of cytochrome *c*. One unit of SOD activity was defifined as the amount of enzyme that inhibits by 50% the rate of cytochrome *c* reduction. The method of Aebi [14] was used to analyze catalase activity. This method uses change in absorbance at 240nm at 25˚C of the solution of 10mM $H_2O_2$ in phosphate buffer, pH 7.0. The decrease in absorbance per unit time is a measure of the catalase activity. Gpx activity was assayed by the method of Paglia and Valentine [15]. The rate of oxidation of the reduced form of glutathione was monitored by a decrease in the concentration of NADPH caused by addition of glutathione reductase to the reaction mixture. All enzyme activities were reported as units/mg of protein.

## 2.8. Immunohistochemical (IHC) Staining

IHC staining was used to assess mesothelial cell healing by cytokeratin-19 (CK-19), alpha-smooth muscle actin (α-SMA) and nuclear factor erythroid 2-related factor 2 (Nrf2). IHC staining was performed using a streptavidin-biotin kit (Maxim, Fuzhou, China) following the manufacturer's instructions. The sections were deparaffinized and rehydrated, incubated with 30 g/L hydrogen peroxide solution at room temperature for 5 minutes, and blocked with goat serum. Subsequently, the sections were incubated with mouse CK-19 (CK-19; 1 : 50 dilution, Abcam, UK), rabbit α-SMA (ab5694, 1 : 100 dilution, Abcam) and Nrf2 antibodies (Nrf2; 1 : 100 dilution, Abcam) at 4˚C overnight. The sections were then incubated with biotinylated rabbit anti-mouse IgG for 20 minutes. Incubation with streptavidin-biotin peroxidase complex at 37˚C was performed for another 20 minutes. The sections were washed in phosphate-buffered saline four times for 5 minutes per wash. Diaminobenzidine tetrahydrochloride was used for visualization, and hematoxylin was used as the counterstain. The sections were dehydrated, mounted, and sealed. To evaluate the expression of these indicators, at least five random high-power fields of adhesion tissue were reviewed for each section as described in the histopathological evaluation. α-SMA staining was used to assess the fibrosis level of tissue in different treatments by using Image-Pro Plus 5.0 software (Leica Qwin. Plus, Leica Microsystem Imaging Solutions Ltd., Cambridge, UK). Then, CK-19 staining was used to assess the rate of the integrity of the mesothelial cells. This rate was calculated by the stained mesothelial cell length in the adhesion tissue divided by the total mesothelial cell length in the selected fields. The Nrf2 scoring system was as follows: 0—no expression, 1—low expression, 2—moderate expression, 3—strong expression, and 4—very strong expression. The expression condition in the adhesive tissue was calculated as the average score of examined sections.

## 2.9. Sirius red picric acid staining

More than 5 pathological sections were randomly selected for Sirius red picric acid staining using the experimental method. Staining was performed with 0.1% Sirius red picric acid (Direct Red 80; Sigma–Aldrich, St. Louis, MO, USA), followed by counterstaining with

hematoxylin. Five high-magnification fields were randomly selected from each pathological section to measure the width of collagen tissue by using Image-Pro Plus 5.0 software. The average thickness of the adhesive area of each tissue was considered the thickness of the adhesive area of each tissue.

## 2.10 Culture of HMrSV5 human peritoneal mesothelial cells

Human HMrSV5 cells were purchased from the American Type Culture Collection (ATCC, Manassas, VA, USA). The cells were cultured in Dulbecco's modified Eagle's medium (DMEM; Gibco, NY, USA) containing 10% fetal bovine serum (FBS; HyClone, Logan, USA) at 37°C in humidified 5% $CO_2$. Transforming growth factor-β1 (TGF-β1) (10 ng/mL, Solarbio Science & Technology Co., Ltd., Beijing, China) was used to induce the epithelial-mesenchymal transition (EMT).

## 2.11 Cellular immunofluorescent staining

HMrSV5 cells ($5\times10^6$) were grown on cover slips. Then, the cells in the AICAR-treated group were cultured in a 1 mg/mL AICAR solution for 2 hours and 5 ng/mL TGF-β1 for 24 hours, while the control group was treated with the same dose of PBS. Next, after washing three times, the cells were fixed with 4% paraformaldehyde overnight. Cellular immunofluorescent staining was performed as described in previous studies. The fluorescence intensity was measured by ImageJ software and compared to the control group.

## 2.12 RNA isolation and real-time PCR

Total RNA was extracted and purified from cells by a kit according to the manufacturer's instructions (Pioneer Biological Co., Ltd. Xi'an, China). Then, the total RNA was reverse-transcribed using SYBR Green (Takara Biotechnology Co., Ltd.) and an ABI 7500 instrument (Thermo Fisher Scientific, Inc.). Primers were designed according to the sequences reported below (Table 1). The real-time PCR data were normalized to endogenous levels and were carried out in triplicate.

## 2.13. Statistical analyses

All data were analyzed with SPSS 22.0 software (Chicago, IL, USA) and are presented as the means ± standard errors of the mean (SEM). Analysis of variance (ANOVA) was performed to determine significant differences in normally distributed data, and the abnormally distributed data were analyzed by the Kruskal–Wallis test. Enumeration data were determined by Fisher's exact test. $p$ values $<0.05$ were considered significant.

**Table 1. Primers designed for different target genes.**

| | | |
|---|---|---|
| Vimentin | Forward | TTGAACGCAAAGTGGAAT |
| | Reverse | AGGTCAGGCTTGGAAACA |
| E-cadherin | Forward | CAACGACCCAACCCAAGAA |
| | Reverse | CCGAAGAAACAGCAAGAGCA |
| α-SMA | Forward | CATCATGCGTCTGGATCTGG |
| | Reverse | GGACAATCTCACGCTCAGCA |
| GADPH | Forward | TGCACCACCAACTGCTTAGC |
| | Reverse | GGCATGGACTGTGGTCATGAG |

## 3. Results

### 3.1 AICAR ameliorates adhesion scores in a rat model

All forty rats finished this experiment, and none died or suffered serious postoperative complications. On the 7th day after surgery, the adhesion scores of all the rats were assessed by Nair's system. All rats in the control group had formed a large number of dense adhesions, while those in the sodium hyaluronate-treated group showed formation of a few thin adhesions. The AICAR-treated group had fewer adhesion formation incidences than the control group (Fig 1A). The adhesion score, which was evaluated by Nair's scoring system, demonstrated that AICAR treatment attenuated the adhesion score compared to that of the control group. However, the nonadhesion rate was not different from that of the control group (Fig 1B and 1C).

### 3.2 AICAR reduces inflammation in adhesion tissue

Inflammation is the initiating cause of adhesion formation. By the HE staining inflammation scoring system, we found that the inflammation score in the AICAR-treated group was dramatically reduced (Fig 2A and 2B). Next, to explore the AICAR-related inflammation in adhension tissue, the level of key proteins involved in inflammation were measured using ELISA and western-blotting analysis (Fig 2C), TGF-β1 concentration was increased in adhesion tissue, while it was significantly decreased in the AICAR-treated group.

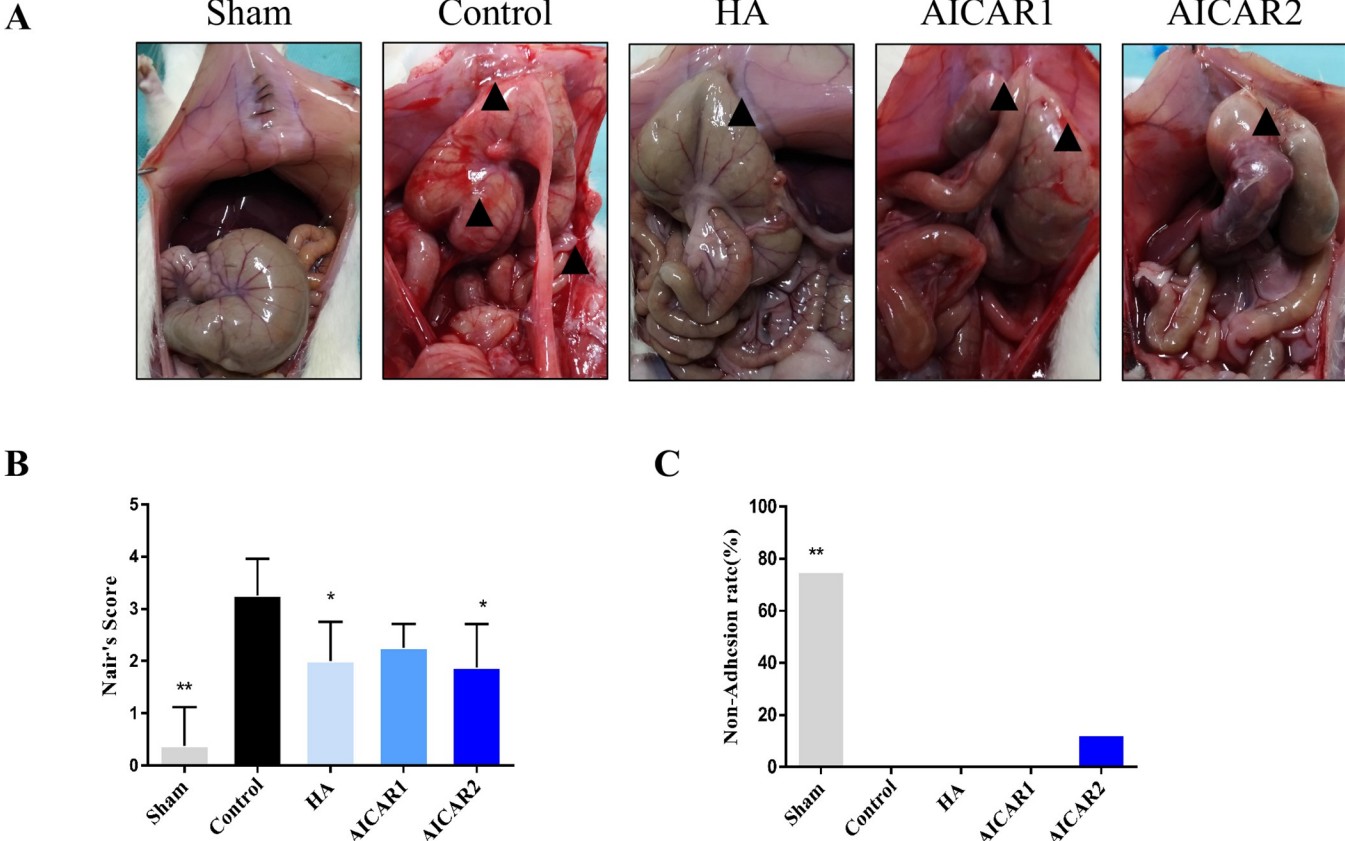

**Fig 1. AICAR ameliorates adhesion scores and reduces inflammation in a rat model (n = 8).** (A) A typical picture of the most representative adhesion situation of each group. The sham group included rats that received only open and close surgery. The control group had massive adhesion, while in the HA- and AICAR-treated groups, adhesion formation was relieved. The black arrow indicates the adhesion bands. (B) The Nair's score of different groups, assessed by Kruskal–Wallis test. (C) The nonadhesion rate in each group. (compared to control, * represents $p < 0.05$, ** represents $p < 0.01$).

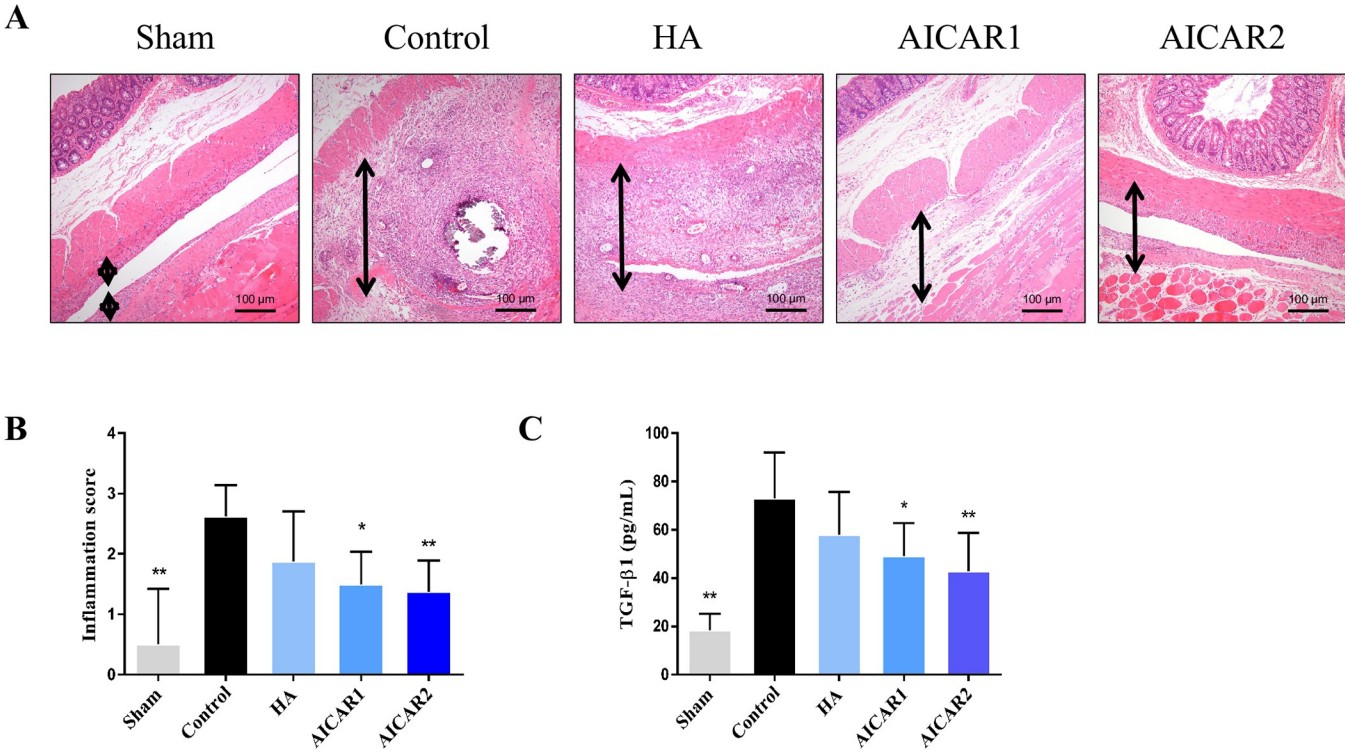

**Fig 2. AICAR reduces inflammation in adhesion tissue.** (A) A typical picture of HE staining from each group. The black arrow indicates the adhesion tissue. (B) The inflammation score in each group; assessed by one-way ANOVA. (C) The TGF-β1 level assessed by ELISA in each group; assessed by Kruskal–Wallis test. (compared to control, * represents $p < 0.05$, ** represents $p < 0.01$).

### 3.3 AICAR alleviates the oxidative stress response in adhesion tissue

ROS play a vital role in the inflammatory reaction of the postoperative adhesion formation process. Here, we found that compared to the control group, the AICAR-treated group had a lower level of ROS and a ROS-related index, MDA ($p<0.05$) (Fig 3A and 3B). However, the NO levels was increased after AICAR treatments ($p<0.05$) (Fig 3C). The activity of SOD, Catalase and Gpx was also significanty decreased in the AICAR-treated group ($p < 0.01$) (Fig 3D). Then, we assessed the Nrf2 levels in adhesion tissue and found that AICAR can promote Nrf2 expression in adhesion tissue ($p<0.05$) (Fig 3E).

### 3.4 AICAR treatment inhibits tissue fibrosis and promotes peritoneal mesothelial cell repair

Adhesion formation is the process of tissue fibrosis. By Sirius red picric acid staining, we found that the AICAR-treated group had a lower fiber thickness than the control group (Fig 4A1 and 4B). CK19 staining demonstrated that the AICAR groups had a higher rate of peritoneal mesenchymal cell integrity than the control (Fig 4A2 and 4C). α-SMA staining showed that the AICAR-treated groups can significantly reduce the fibrogenesis in vivo (Fig 4A3 and 4D), but the mechanism is still unclear.

Peritoneal mesothelial cell mesenchymal transition (MMT) is an important process in fibrosis. Here, in a TGF-β1-induced HMrSV5 human cell MMT model, we found that AICAR treatment can reduced α-SMA and vimentin mRNA expression compared to that in the TGF-

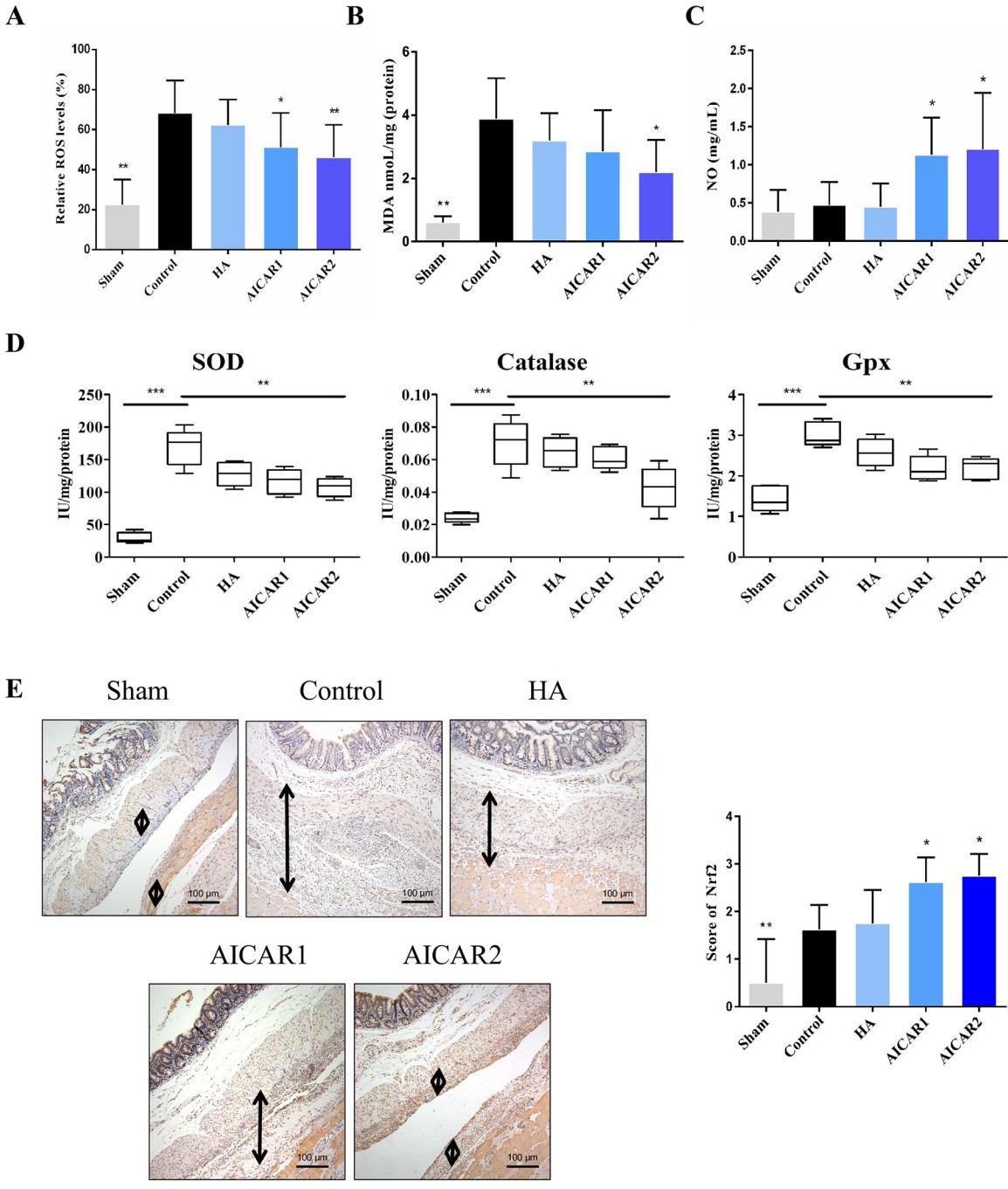

**Fig 3. AICAR alleviates the oxidative stress response in adhesion tissues.** (A) The Relative ROS levels in each group, assessed by Kruskal–Wallis test. (B) The MDA level in each group, assessed Kruskal–Wallis test. (C) The NO levels in each group, assessed Kruskal–Wallis test. (D) The SOD, catalase, Gpx activity in each group; assessed by Kruskal–Wallis test. (E) Typical picture of Nrf2 IHC staining from each group. The black arrow indicates the adhesion tissue. The IHC score in each group, assessed by one-way ANOVA. (compared to control, * represents $p < 0.05$, ** represents $p < 0.01$).

β1 group, while the E-cadherin mRNA expression was higher in the AICAR group (Fig 5A–5C). Cell immunofluorescence also showed that the α-SMA fluorescence intensity was down-regulated in the AICAR group compared to it in the TGF-β1 group (Fig 5D).

**A**

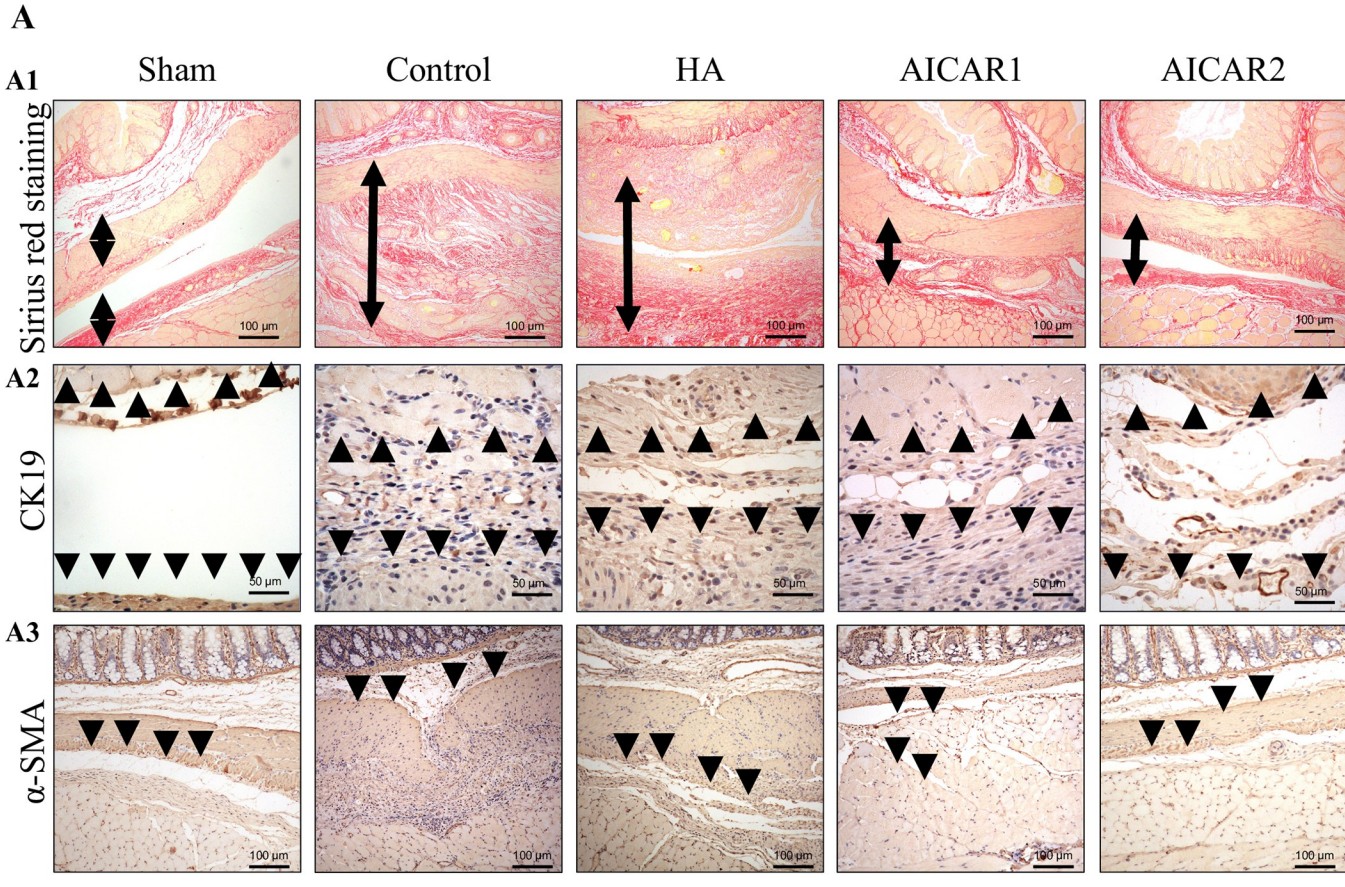

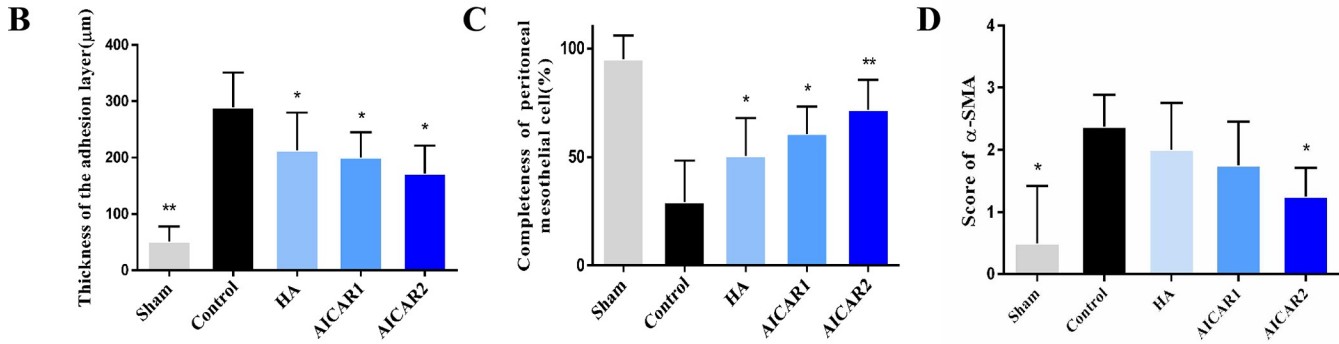

**Fig 4. AICAR treatment inhibits tissue fibrosis and promotes peritoneal mesothelial cell repair *in vivo*.** (A1) A typical picture of Sirius Red picric acid staining from each group. (A2) shows CK19 IHC staining from each group. The black arrow indicates the adhesion tissue or normal peritoneal tissues. (A3) α-SMA IHC staining from each group. The black arrow indicates the α-SMA staining. (B) The fiber thickness in each group was measured by Sirius Red picric acid staining, assessed by Kruskal–Wallis test. (C) The integrity of peritoneal mesothelial cells in each group, assessed Kruskal–Wallis test. (D) The tissue fibrogenesis in each group, assessed Kruskal–Wallis test. (compared to control, $*$ represents $p < 0.05$, $**$ represents $p < 0.01$).

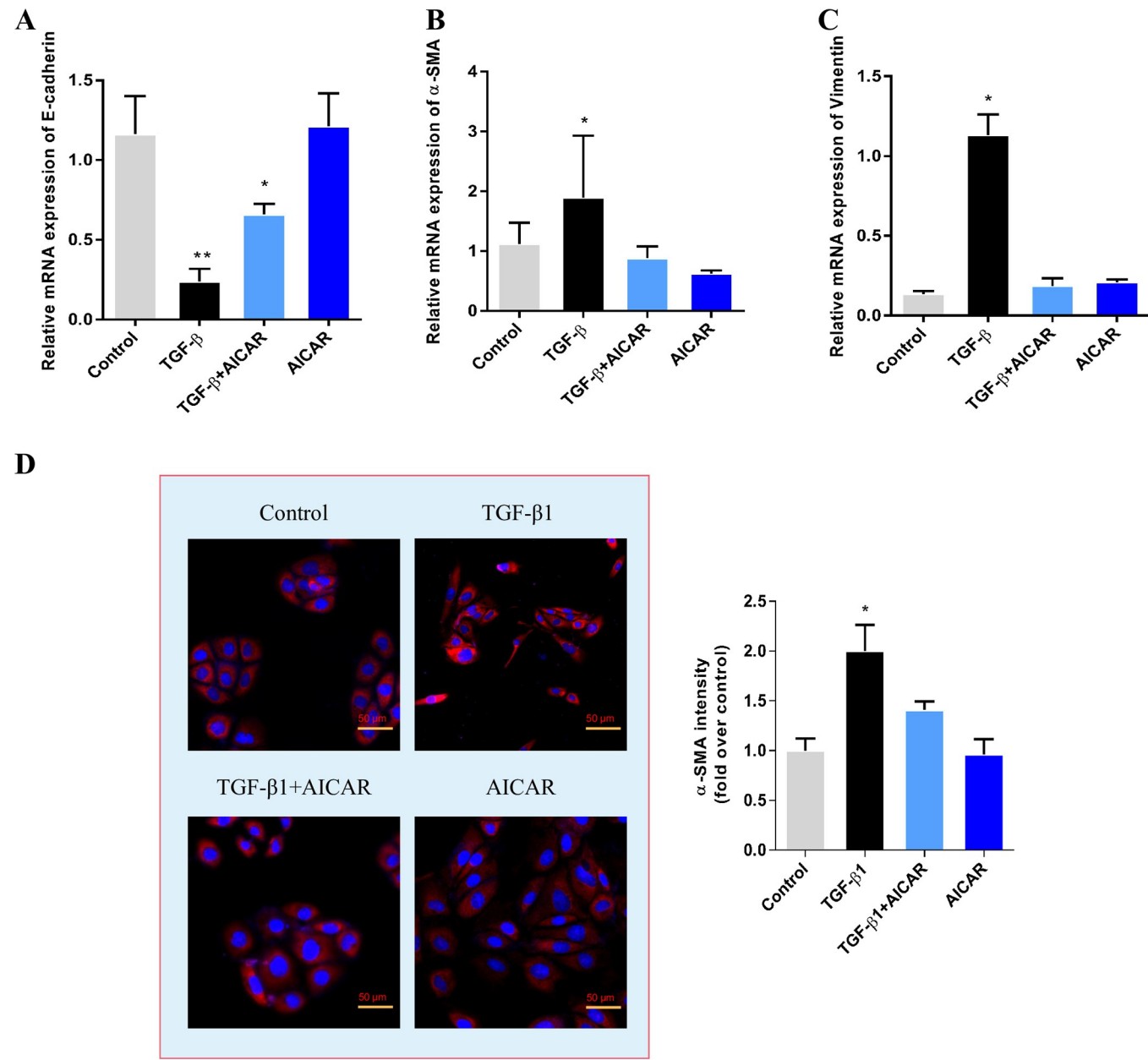

**Fig 5. AICAR treatment inhibits mesothelial cell mesenchymal transition *in vitro*.** q-PCR was used to assess E-cadherin mRNA expression in each group, n = 3, Kruskal–Wallis test. (B) q-PCR was used to assess α-SMA mRNA expression in each group, n = 3, Kruskal–Wallis test. (C) q-PCR was used to assess vimentin mRNA expression in each group, n = 3, Kruskal–Wallis test. (D) Cellular immunofluorescence staining was used to assess α-SMA expression in each group. Quantification located in adjacent graph. n = 3, Kruskal–Wallis test. (compared to control, * represents $p < 0.05$, ** represents $p < 0.01$).

## 4. Discussion

The AMPK pathway is an important pathway to maintain energy homeostasis in our body that participates widely in stress (ischemic, oxidative, hypoglycemic), exercise, and hormone (adiponectin, leptin) pathways and can be modulated by pharmacological agents [16]. Here, we demonstrated that AMPK agonists can alleviate inflammatory reactions, decrease oxidative stress response and promote mesothelial cell repair in postoperative adhesion formation.

These results demonstrated that promoting the AMPK pathway may prevent adhesion formation.

As an AMPK pathway agonist, AICAR can reduce inflammatory reactions in many disease [17]. AICAR can inhibit NF-kB DNA binding independently of AMPK to attenuate lipopoly-saccharide (LPS) triggered inflammatory responses in human macrophages [18]. AICAR can also inhibit NF-kB activation and reduce the expression of interleukin-1β (IL-1β) in inflammatory pain [19]. In an intestinal ischemia and reperfusion model, AICAR treatment has also been shown to decrease the inflammatory response and alleviate tissue damage caused by ischemia [20]. Here, we found by HE staining and TGF-β1 assessment that AICAR can inhibit the inflammation score in adhesion tissue and that the level of inflammatory factor TGF-β1 was also reduced by AICAR treatment.

During the formation of postoperative abdominal adhesions, due to the destruction of local tissues, a large amount of reactive oxygen species is generated, which in turn aggravates the inflammatory response and leads to tissue damage [21]. AICAR can decrease sepsis or LPS-induced endothelial activation and organ injury by reducing oxidative stress response [22]. Here, we found that AICAR may play a citical role in anti-oxidative effects, it can reduce ROS, SOD, catalase and Gpx activity in adhesion tissue. And it can also promote the expression of Nrf2. Nrf2 is best known as an oxidant stress response transcription factor, it lies at the center of a complex regulatory network and has a mutiple function. The most function exerted by Nrf2 is the protective role in anti-inflammation conditions and promoting cells repairment [23]. The peritoneum is composed of peritoneal mesothelial cells and the underlying connective tissue [24]. After damage, peritoneal mesothelial cells are missing. The peritoneal cells that repair the local tissue are derived from the surrounding normal peritoneal cells or other stem cells. However, in an inflammatory environment, these cells can undergo necrosis or become fibroblasts. This transformation will promote the formation of a local adhesion, and thus inhibition of peritoneal mesothelial cell-to-interstitial transformation is very important for the prevention and treatment of postoperative abdominal adhesion [25, 26]. In the *in vivo* experiment, we found that AICAR can reduce collagen deposition and fibrogenesis during adhesion and promote mesothelial cell repair. To further understand its possible mechanism, we extracted and cultured rat primary peritoneal mesothelial cells, and the results showed that AICAR treatment can significantly inhibit TGF-β1-induced peritoneal mesothelial cell-to-mesenchymal transition.

In conclusion, our study demonstrated that an AMPK pathway agonist can inhibit postoperative adhesion formation by reducing inflammation, decreasing oxidative stress response and promoting peritoneal mesothelial cell repair. This finding indicated that AICAR may be used as a potential drug to prevent the formation of postoperative abdominal adhesions.

## Supporting information

**S1 Dataset.**
(RAR)

## Acknowledgments

We thank the central Laboratory of Shaanxi Provincial People's Hospital for providing molecular biotechnology support. We also thank American Journal Experts (AJE) for language polishing services.

## Author Contributions

**Conceptualization:** Fei Xue.

**Data curation:** Yunhua Wu, Zengzhan Gao, Fei Xue.

**Formal analysis:** Yunhua Wu, Zengzhan Gao, Ni Yang.

**Funding acquisition:** Yunhua Wu, Ni Yang, Fei Xue.

**Investigation:** Yunhua Wu, Ni Yang.

**Methodology:** Yunhua Wu.

**Project administration:** Yunhua Wu, Xianglong Duan, Fei Xue.

**Resources:** Yunhua Wu, Xianglong Duan, Zengzhan Gao.

**Software:** Fei Xue.

**Supervision:** Yunhua Wu, Xianglong Duan, Fei Xue.

**Writing – original draft:** Yunhua Wu, Fei Xue.

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
