## [Decision Letter · Decision Letter 0]

9 Jun 2022

PONE-D-22-09036AICAR attenuates postoperative abdominal adhesion formation by inhibiting oxidative stress and promoting mesothelial cell repairPLOS ONE

Dear Dr. Xue,

Thank you for submitting your manuscript to PLOS ONE. After careful consideration, we feel that it has merit but does not fully meet PLOS ONE’s publication criteria as it currently stands. Therefore, we invite you to submit a revised version of the manuscript that addresses the points raised during the review process.

We look forward to receiving your revised manuscript.

Kind regards,

Md Ekhtear Hossain, Ph.D.

Academic Editor

PLOS ONE

Journal Requirements:

2. As part of your revision, please complete and submit a copy of the Full ARRIVE 2.0 Guidelines checklist, a document that aims to improve experimental reporting and reproducibility of animal studies for purposes of post-publication data analysis and reproducibility: https://arriveguidelines.org/sites/arrive/files/Author%20Checklist%20-%20Full.pdf (PDF). Please include your completed checklist as a Supporting Information file. Note that if your paper is accepted for publication, this checklist will be published as part of your article.

“This study was supported by 2021 Science and technology Talents Support project of Shaanxi Provincial People's Hospital (2021JY-08) and (2021JY-15).”

“Fei Xue received 2021 Science and technology Talents Support project of Shaanxi Provincial People's Hospital (2021JY-08), Yunhua Wu received 2021 Science and technology Talents Support project of Shaanxi Provincial People's Hospital (2021JY-15),The funders had no role in study design, data collection and analysis, decision to publish, or preparation of the manuscript.”

Reviewers' comments:

Reviewer's Responses to Questions

**Comments to the Author**

1. Is the manuscript technically sound, and do the data support the conclusions?

Reviewer #1: Yes

Reviewer #2: Yes

2. Has the statistical analysis been performed appropriately and rigorously? 

Reviewer #1: Yes

Reviewer #2: Yes

3. Have the authors made all data underlying the findings in their manuscript fully available?

Reviewer #1: No

Reviewer #2: Yes

4. Is the manuscript presented in an intelligible fashion and written in standard English?

Reviewer #1: Yes

Reviewer #2: Yes

5. Review Comments to the Author

Reviewer #1: In this manuscript the authors investigated the role of AMPK agonist AICAR in inhibiting post-operative abdominal adhesion formation. The authors showed that AICAR can reduce adhesion formation by inhibiting oxidative stress and facilitate peritoneal repair in a rat model.

Here’s my comments on the manuscript:

Major Comments:

1.Multiple studies indicated that p-eNOS expression and nitric oxide (NO) production were increased by in vivo administration of AICAR; please include results of NO and p-eNOS levels in your in vivo study groups. This data will further justify the effect of AICAR on oxidative stress.

2.As the authors indicated in the discussion section, “Peritoneal mesothelial cell mesenchymal transition (MMT) is an important process in fibrosis” please include the results of α-SMA and vimentin expression levels in your in vivo study groups (like you did for Nrf2).

3.In Figure 5D, the number of visible cells in the TGF-β group is way less than in other groups, which dismantled the validity of the quantitative data; please replace this image with the better one and recalculate the graph accordingly. 

Minor comments:

1.Indicate the full name of AICIR (5-aminoimidazole-4-carboxyamide ribonucleoside) in the Material and Methods section and the purchasing information (company, purity, etc.).

2.Please include brief information about Nair’s scoring system and add a reference.

3.Please includes the purchasing information for the MDA detection kit.

4.In Figure 5D, the authors measured the transcriptional (mRNA) levels of E-cadherin, a-SMA, and Vimentin. Please indicate it in the figure, e.g., Relative mRNA expression of E-cadherin.

5.Please italicize the words in vivo and in vitro in the manuscript. The statistically significant symbol “P” should always indicate by a small letter (and italicize it); please correct it in the manuscript.

6.In Figure 3A, please replace “Relative ROS expression (%)” with “Relative ROS levels (%)” and subsequently correct the figure legend.

Reviewer #2: The manuscript by Xue F et al presented data linking AMPK agonist AICAR with post-operative abdominal adhesion formation. The experimental design was well constructed and showed the protective effect of AICAR on adhesion formation by inhibiting oxidative stress and peritoneal repair in the post-operative rats. Although the results are interesting, the reviewer has couple of comments.

1. In the abstract-back ground section should describe solely context of the research rather than describing results and concluding remark.

2. The authors conclude that AMPK activator AICAR inhibits inflammation and oxidative stress in the adhesion tissue. Thus, the author should first confirm AMPK and inflammatory signaling (i.e TGF-β and NF-kb) activation in the adhesion tissue by western blotting. All they evaluated to confirm anti-oxidative effects of AICAR were ROS and MDA. Thus reviewer is interested to see Sod, catalase and Gpx expression in the adhesion tissue.

3. They have shown inhibitory effects of AICAR on fibrosis and adhesion formation. Therefore, it would be interesting to see the protein abundance of collagen1, α-SMA, integrins and E-cadherin in the adhesion tissue of the post-operative rats.

6. PLOS authors have the option to publish the peer review history of their article (what does this mean?). If published, this will include your full peer review and any attached files.

Reviewer #1: No

Reviewer #2: No

---

## [Author Response · Author response to Decision Letter 0]

26 Jul 2022

Reviewer #1 (Reviewer Comments to the Author)

Response to Reviewer #1

Many thanks for your expert comments and suggestions for improving our manuscript. We have followed your advice and have improved the presentation of the manuscript accordingly.

1.Multiple studies indicated that p-eNOS expression and nitric oxide (NO) production were increased by in vivo administration of AICAR; please include results of NO and p-eNOS levels in your in vivo study groups. This data will further justify the effect of AICAR on oxidative stress.

Response: Thank you very much for your advice, to further demonstrate the effect of AICAR on the expression NO and p-eNOS, we had both detected the NO level by ELISA and the p-eNOS by IHC, however, we had fail the for the IHC staining of p-eNOS. However, the NO level had been increased by the AICAR treatments. As shown in Figure 3C.

2.As the authors indicated in the discussion section, “Peritoneal mesothelial cell mesenchymal transition (MMT) is an important process in fibrosis” please include the results of α-SMA and vimentin expression levels in your in vivo study groups (like you did for Nrf2).

Response: Thank you very much for your advice, we have added the α-SMA IHC staining for each group, as shown in figure (4A3 and 4D). The result showed that AICAR treatment can protect the integrity of peritoneal mesenchymal cell and inhibits tissue fibrogenesis. Due to a lack of time, we did not preformed the vimentin IHC staining.

3.In Figure 5D, the number of visible cells in the TGF-β group is way less than in other groups, which dismantled the validity of the quantitative data; please replace this image with the better one and recalculate the graph accordingly. 

Response: Thank you very much. We chose fewer cell image fields in order to have a better view of cell morphological changes and fluorescence changes. Since the peritoneal mesothelial cells had turned into fibers and became smaller, they appeared to have fewer cells, so we have replaced with another one.

Reviewer #2 (Reviewer Comments to the Author)

Response to Reviewer #2

Many thanks for your expert comments and suggestions for improving our manuscript. We have followed your advice and have improved the presentation of the manuscript accordingly.

1.Indicate the full name of AICIR (5-aminoimidazole-4-carboxyamide ribonucleoside) in the Material and Methods section and the purchasing information (company, purity, etc.).

Response: Thank you for for carefully reminding, we purchased the AICIR from cell signaling technology, USA, purity >99%.

2.Please include brief information about Nair’s scoring system and add a reference.

Response: Thank you very much, we have added the Nair’s scoring system in our manuscript and added the citation.

3.Please includes the purchasing information for the MDA detection kit.

Response: Thank you for your advice, we have added the related information in our manuscript.

4.In Figure 5D, the authors measured the transcriptional (mRNA) levels of E-cadherin, a-SMA, and Vimentin. Please indicate it in the figure, e.g., Relative mRNA expression of E-cadherin.

Response: Thank you for your reminding, we have revised them.

5.Please italicize the words in vivo and in vitro in the manuscript. The statistically significant symbol “P” should always indicate by a small letter (and italicize it); please correct it in the manuscript.

Response: Thank you very much for your reminding, we have revised them.

6.In Figure 3A, please replace “Relative ROS expression (%)” with “Relative ROS levels (%)” and subsequently correct the figure legend.

Response: Thank you very much for your reminding, we have revised them.

Reviewer #3 (Reviewer Comments to the Author)

Response to Reviewer #3

1.In the abstract-back ground section should describe solely context of the research rather than describing results and concluding remark.

Response: Thank you for your suggestion, we have revised it.

2.The authors conclude that AMPK activator AICAR inhibits inflammation and oxidative stress in the adhesion tissue. Thus, the author should first confirm AMPK and inflammatory signaling (i.e TGF-β and NF-kb) activation in the adhesion tissue by western blotting. All they evaluated to confirm anti-oxidative effects of AICAR were ROS and MDA. Thus reviewer is interested to see Sod, catalase and Gpx expression in the adhesion tissue.

Response: Thank you for your suggestion. We have shown the TGF-β1 concentration result after different treatments in Figure 2C, and we can also show you the time-course result we did to confirm that TGF-β1 accumulation in adhesion tissue. Please see in below. But for this part of result, we don’t want to added in the figures. 

Since NF-kb signaling pathway activation has been widely confirmed in adhesion tissue[1-4]. So, we don’t think we need to repeat this part again. 

1.Chen S, Jiang S, Zheng W, Tu B, Liu S, Ruan H, Fan C. RelA/p65 inhibition prevents tendon adhesion by modulating inflammation, cell proliferation, and apoptosis. Cell Death Dis. 2017 Mar 30;8(3):e2710. doi: 10.1038/cddis.2017.135. PMID: 28358376; PMCID: PMC5386538.

2.Yang L, Lian Z, Zhang B, Li Z, Zeng L, Li W, Bian Y. Effect of ligustrazine nanoparticles on Th1/Th2 balance by TLR4/MyD88/NF-κB pathway in rats with postoperative peritoneal adhesion. BMC Surg. 2021 Apr 26;21(1):211. doi: 10.1186/s12893-021-01201-7. PMID: 33902534; PMCID: PMC8077798.

3.Jiang ZL, Fletcher NM, Diamond MP, Abu-Soud HM, Saed GM. Hypoxia regulates iNOS expression in human normal peritoneal and adhesion fibroblasts through nuclear factor kappa B activation mechanism. Fertil Steril. 2009 Feb;91(2):616-21. doi: 10.1016/j.fertnstert.2007.11.059. Epub 2008 Feb 20. PMID: 18281043; PMCID: PMC2812021.

4.Li L, Frei B. Iron chelation inhibits NF-kappaB-mediated adhesion molecule expression by inhibiting p22(phox) protein expression and NADPH oxidase activity. Arterioscler Thromb Vasc Biol. 2006 Dec;26(12):2638-43. doi: 10.1161/01.ATV.0000245820.34238.da. Epub 2006 Sep 14. PMID: 16973969.

For Sod, catalase and Gpx expression, we have added Sod, catalase and Gpx activity analysis in Figure 3D. 

3.They have shown inhibitory effects of AICAR on fibrosis and adhesion formation. Therefore, it would be interesting to see the protein abundance of collagen1, α-SMA, integrins and E-cadherin in the adhesion tissue of the post-operative rats.

Response: Thank you for your advice, we have added the α-SMA IHC staining of post-operative rats in our manuscript, as shown in figure (4A3 and 4D).

---

## [Editor Report · Decision Letter 1]

29 Jul 2022

AICAR attenuates postoperative abdominal adhesion formation by inhibiting oxidative stress and promoting mesothelial cell repair

PONE-D-22-09036R1

Dear Dr. Xue,

We’re pleased to inform you that your revised manuscript has been judged scientifically suitable for publication and will be formally accepted for publication once it meets all outstanding technical requirements.

Kind regards,

Md Ekhtear Hossain, Ph.D.

Academic Editor

PLOS ONE

---

## [Editor Report · Acceptance letter]

4 Aug 2022

PONE-D-22-09036R1 

AICAR attenuates postoperative abdominal adhesion formation by inhibiting oxidative stress and promoting mesothelial cell repair 

Dear Dr. Xue:

I'm pleased to inform you that your manuscript has been deemed suitable for publication in PLOS ONE. Congratulations! Your manuscript is now with our production department. 

Kind regards, 

on behalf of

Dr. Md Ekhtear Hossain 

Academic Editor

PLOS ONE